# Identification of Vaginal Microbial Communities Associated with Extreme Cervical Shortening in Pregnant Women

**DOI:** 10.3390/jcm9113621

**Published:** 2020-11-10

**Authors:** Monica Di Paola, Viola Seravalli, Sara Paccosi, Carlotta Linari, Astrid Parenti, Carlotta De Filippo, Michele Tanturli, Francesco Vitali, Maria Gabriella Torcia, Mariarosaria Di Tommaso

**Affiliations:** 1Department of Biology, University of Florence, Sesto Fiorentino, 50019 Florence, Italy; monica.dipaola@unifi.it; 2Department of Health Sciences, Division of Obstetrics & Gynecology, University of Florence, 50139 Florence, Italy; viola.seravalli@unifi.it (V.S.); carlotta.linari@unifi.it (C.L.); mariarosaria.ditommaso@unifi.it (M.D.T.); sara.paccosi@unifi.it (S.P.); astrid.parenti@unifi.it (A.P.); 3Institute of Agricultural Biology and Biotechnology, National Research Council, 56124 Pisa, Italy; carlotta.defilippo@ibba.cnr.it (C.D.F.); francesco.vitali@ibba.cnr.it (F.V.); 4Department of Experimental and Clinical Medicine, University of Florence, 50139 Florence, Italy; michele.tanturli@unifi.it

**Keywords:** high-risk pregnancy, shortened cervix, microbiome, *Lactobacillus*

## Abstract

The vaginal microbiota plays a critical role in pregnancy. Bacteria from *Lactobacillus* spp. are thought to maintain immune homeostasis and modulate the inflammatory responses against pathogens implicated in cervical shortening, one of the risk factors for spontaneous preterm birth. We studied vaginal microbiota in 46 pregnant women of predominantly Caucasian ethnicity diagnosed with short cervix (<25 mm), and identified microbial communities associated with extreme cervical shortening (≤10 mm). Vaginal microbiota was defined by 16S rRNA gene sequencing and clustered into community state types (CSTs), based on dominance or depletion of *Lactobacillus* spp. No correlation between CSTs distribution and maternal age or gestational age was revealed. CST-IV, dominated by aerobic and anaerobic bacteria different than *Lactobacilli*, was associated with extreme cervical shortening (odds ratio (OR) = 15.0, 95% confidence interval (CI) = 1.56–14.21; *p* = 0.019). CST-III (*L. iners*-dominated) was also associated with extreme cervical shortening (OR = 6.4, 95% CI = 1.32–31.03; *p* = 0.02). Gestational diabetes mellitus (GDM) was diagnosed in 10/46 women. Bacterial richness was significantly higher in women experiencing this metabolic disorder, but no association with cervical shortening was revealed by statistical analysis. Our study confirms that *Lactobacillus*-depleted microbiota is significantly associated with an extremely short cervix in women of predominantly Caucasian ethnicity, and also suggests an association between *L. iners*-dominated microbiota (CST III) and cervical shortening.

## 1. Introduction

The uterine cervix acts as a physical and immune barrier against pathogens’ passage into the uterine cavity during pregnancy. Premature cervical remodeling, shortening, and dilation of the cervix are known risk factors for spontaneous preterm birth (sPTB) [1,2,3,4] with the notion that the shorter the cervix, the higher the risk of sPTB [1]. In addition to congenital disorders [5], genetic syndromes (e.g., Ehlers–Danlos syndrome) [6] and progesterone deficiency [7], local inflammation secondary to changes in the cervico-vaginal microbiome is another mechanism that has been proposed to cause cervical shortening [8].

Vaginal microbial communities are largely involved in preventing ascending infections from the vagina into the uterine cavity by pathogens that can seriously compromise pregnancy [9,10]. At least five microbial communities, referred to as community state types (CSTs), have been identified in the vaginal microbiota of healthy reproductive-age non-pregnant women. Four CSTs are dominated by *Lactobacillus* species, better adapted to the vaginal environment [11]. In particular, CST-I is dominated by *Lactobacillus crispatus*, CST-II by *L. gasseri*, CST-III by *L. iners*, CST-V by *L. jensenii*. Each species contributes to the first-line defense against bacterial, fungal, and viral pathogens through the release of antimicrobial and anti-inflammatory products and the production of lactic acid that maintains a low vaginal pH [10,12].

CST-IV is represented by polymicrobial communities that include species belonging to *Gardnerella, Atopobium, Mobiluncus, Megasphoera Prevotella, Streptococcus, Mycoplasma, Ureaplasma, Dialister,* and *Bacteroides* genera [11,13]. CST-IV is more common in African, African-American, and Hispanic women, being detected in 40% to 60% of non-pregnant black women, depending on the country analyzed [10,12,14]. In contrast, the prevalence of CST-IV in Caucasian women is around 10%, as reported in cohorts of non-pregnant women [12,14,15]. High stability of the *Lactobacillus* community was recorded during pregnancy [16,17]. Some reports indicated that the vaginal microbiota is more stable during pregnancy than in non-pregnant women [17].

Recent results showed that the vaginal microbiota depleted by *Lactobacillus* spp. (CST-IV) is associated with increased odds of having a short cervix and, that the concomitance of both CST-IV and a short cervix increases the risk of sPTB [8,18]. These results were obtained in a large cohort of pregnant women, mainly of Hispanic, African, African-American ethnicity, whose vaginal microbiota in non-pregnant status is most frequently of the CST-IV type with few to no *Lactobacillus* spp. detected [11,14].

Moreover, metabolic disorders during pregnancy, such as gestational diabetes mellitus (GDM), are known to affect the composition of the vaginal microbiota and the immune homeostasis of the vaginal environment, by being associated with an abundance of potential pathogens and an increase of inflammatory cytokines [19].

In this study, we recruited a cohort of pregnant women (of whom 95% were of Caucasian ethnicity) with sonographic evidence of cervical shortening (<25 mm) revealed in the second or early third trimester of pregnancy. In particular, we aimed to characterize microbial communities associated with cervical length ≤10 mm, which we defined as “extreme cervical shortening”, as it has been associated with higher rates of PTB compared to a longer measurement of the cervix (e.g., 20 or 25 mm), at different gestational ages [1,20]. The influence of GDM on vaginal microbiome composition and cervical shortening was also studied.

## 2. Experimental Section

### 2.1. Study Population and Sample Collection

The study population consisted of asymptomatic pregnant women with singleton gestation who were referred to the Preterm Birth Clinic of the Department of Obstetrics and Gynecology of Careggi University Hospital (Florence, Italy) between 2014 and 2018 for a cervical length <25 mm in the second or early third trimester (23–32 weeks’ gestation). The study was approved by the Ethical Committee of Azienda Ospedaliero-Universitaria Careggi, Firenze (Ref. no. BIO14.0009- 09/07/2014), and all women provided written informed consent. The patients were referred by their obstetricians who detected cervical shortening on a transvaginal ultrasound performed during a routine prenatal visit. Although not under a specific protocol, in private practice in Italy pregnant women are often offered cervical length measurement and this can lead to a diagnosis of a short cervix even after 24–25 weeks of gestation. In some cases, particularly for women who had a cervical length <10 mm, it was the detection of a shortened cervix on a vaginal exam that justified the ultrasound measurement of the cervix.

Exclusion criteria were a history of sPTB, previous surgery to the cervix (cone biopsy and large loop excision), evidence of premature rupture of membranes or symptomatic uterine contractions at the time of recruitment, the presence of fetal abnormalities, vaginal symptoms consistent with infection at the time of enrolment, and the presence of a cervical cerclage or pessary in place at the time of enrolment. All women who are referred to our preterm birth clinic undergo a full obstetric exam and transvaginal ultrasound for assessment of cervical length and a complete medical history is collected by the attending physician. For the purpose of this study, only patients with a singleton gestation and short cervix who did not present any of the exclusion criteria were offered to participate in the study.

Gestational age (GA) was calculated based on the last menstrual period and confirmed by ultrasound. Clinical and demographic information and obstetric history were collected from patients’ charts. Repeat measurement of the cervical length by transvaginal ultrasound was performed by trained personnel to confirm cervical shortening at the time of study recruitment. All patients underwent a complete clinical and vaginal examination. Vaginal secretions were collected by inserting a swab approximately 4 to 5 cm into the vagina and gently rotating it several times. The swab was then placed in phosphate-buffered saline (PBS) on ice for 30 min. After swab removal, samples were centrifuged at 8000× *g* for 10 min, and pellet and supernatant were separately collected and stored at −80 °C. Vaginal progesterone, at a dose of 200 mg daily, was prescribed to 33 patients after sample collection and continued until 34 weeks or until delivery. Placement of a cerclage to prevent PTB in our cohort of patients was not indicated, as “ultrasound-indicated” cerclage in women with short cervix is only considered effective in patients with a history of prior sPTB, which none of our patients had. In addition, none of the patients enrolled in the study received a pessary for prevention of preterm birth after enrollment, either because they did not meet the criteria for pessary placement (cervical length <20 mm before 24 weeks based on our hospital protocol), or because they refused it. On the other hand, women who already had a pessary in place and were referred to the preterm birth clinic for follow-up after discharge from the hospital were considered ineligible for the study, as the impact that the pessary might have on the vaginal microbiota is unknown. All patients were followed until delivery in our preterm birth clinic. Pregnancy and delivery outcomes were collected from patients’ charts.

### 2.2. Bacterial DNA Extraction, 16S Ribosomal RNA Gene Amplicon Preparation, and Illumina MiSeq Sequencing

Bacterial genomic DNA was extracted from the thawed vaginal samples by using the QIAamp DNA Mini Kit (Qiagen, Milano, Italy), following the manufacturer’s protocol. Quality control was carried out by gel electrophoresis and measuring ng/μL of DNA by using Qubit 4 Fluorometer (Thermo Fisher Scientific, Milan, Italy) and the related Qubit dsDNA HS (High Sensitivity) assay kit highly selective for double-stranded DNA(Thermo Fisher Scientific, Italy). The library of 16S rRNA gene amplicons was prepared by IGA Technology Services (Udine, Italy) through amplification of the V3-V4 hypervariable region by using specific-barcoded primers with overhanging adapters. The standard protocol was followed according to the 16metagenomic sequencing library preparation guide from Illumina (Part # 15044223 Rev. B; https://support.illumina.com/). Pooled V3-V4 amplicon libraries were sequenced using the Illumina MiSeq platform.

### 2.3. Sequencing Analysis

The 300-bp paired-end reads obtained from Illumina MiSeq platform for each sample were demultiplexed and quality checked using FastQC 0.11.5. Reads were further processed using the MICCA pipeline (version 1.7.2, http://compmetagen.github.io/micca/) [21] for merging and filtering of reads, chimera checking, and picking of operational taxonomic unit (OTU)/sequence variant (SV), as reported by Meriggi et al. [22]. We obtained 5,705,871 total read counts, with an average per sample equal to 124,040 ± 47,905 (mean ± standard deviation (SD)). Sequence data are available at http://www.ebi.ac.uk/ena/data/view/PRJEB37121, under the accession number PRJEB37121.

### 2.4. Microbiota Data Analysis

Sequence data analyses were performed in R (v.3.42; R Core Team, 2018), by using phyloseq package v.1.22.3 [23]. Alpha diversity analysis and principal coordinate analysis (PCoA) ordination (beta diversity) based on the Bray–Curtis distances and plots for microbial profile comparison among samples were performed by using the Microbiome Analyst tool (https://www.microbiomeanalyst.ca/) [24], providing OTUs and taxonomy tables and the metadata file. Data filtering was set to remove low-quality features. Considering a total of 1182 OTUs number and 1082 OTUs with ≥2 counts, we fixed the low count filter: minimum count (*n* = 2) and 20% prevalence in samples. Low variance filter: 10% of features removed based on standard deviation. Count data were scaled based on cumulative sum scaling (CSS).

*Lactobacillus* spp. were assigned based on the Basic Local Alignment Search Tool nucleotide (BLASTn) software (National Center for Biotechnology Information-NCBI database). The highest percentage of identity (Query cover 100% or 99% and Identity 99 or 95%) and expectation value (E-value) was considered to select significant BLAST hits, keeping only outcomes with the lowest E-value (minimal E-value of 10^−3^). Based on the *Lactobacillus* spp. assignment, Community State Types (CSTs) were defined considering the relative abundance of *Lactobacillus* spp. (>60% in each sample), and aerobic and anaerobic bacteria (ranging from 14 to 40%), as previously indicated [13].

Permutational multivariate analysis of variance (PERMANOVA) and diversity indices calculation (OTU, Shannon, Chao), were calculated using the Microbiome Analyst tool (https://www.microbiomeanalyst.ca/) [24].

To discover potential microbial biomarkers with statistical significance the linear discriminant analysis (LDA) effect size (LEfSe) method was assessed. An alpha significance level of 0.05, either for the factorial Kruskal–Wallis test among classes or for the pairwise Wilcoxon test between subclasses, was used. A size-effect threshold of 2.0 on the logarithmic LDA score was applied for discriminative microbial biomarkers.

### 2.5. Matrix Metalloproteinase-8 Concentration Measurement

From vaginal fluids, protein concentration was assessed by BCA assay (Euroclone, Milan, Italy). A total of 60 micrograms of proteins was suspended in PBS and matrix metalloproteinase-8 (MMP-8) concentration was determined by Immunoplex assay (Millipore, Burlington, MA, USA) using Bioplex instrument [25].

### 2.6. Statistical Analysis

For continuous variables, multiple group comparisons were performed by the Kruskal–Wallis test, while for two-group comparisons the Mann–Whitney U test was used. Spearman’s rank correlation test was used to examine the relationships between two continuous variables. Analysis of covariance (ANCOVA) was applied to compare cervix length among CSTs, obstetric diseases, and gestational diabetes mellitus groups, adjusting for gestational age at sampling or for age of patients. For the analysis of frequency, statistical analyses were performed using Fisher’s exact test. For each odds ratio, a confidence interval at 95% (CI 95%) was shown, and the z test was applied to obtain a *p*-value associated with OR. *p* < 0.05 was considered significant. Statistical analysis was performed using R software version 4.0.2.

## 3. Results

### 3.1. Population

During the study period, 174 potential study participants who met the inclusion criteria were identified. Of these, 128 were excluded because they presented one or more of the exclusion criteria. Therefore, 46 women were enrolled in the study. Table 1 summarizes demographic and clinical information of the enrolled patients, including age (34.2 ± 6.7 years; mean ± SD), gestational age (GA) at sampling (27.6 ± 2; mean ± SD), and pregnancy complications including GDM (diagnosed in 10/46 pregnant women, and requiring insulin treatment in 5 of them). GDM was diagnosed using the 2 h 75 g oral glucose tolerance test at 24 to 28 weeks’ gestation, or earlier in patients at high risk for gestational diabetes. The mean gestational age at diagnosis was 25.1 weeks. Vaginal infections that occurred later in gestation were also reported. In our cohort of women, those enrolled at 28 to 32 weeks all had a cervical length <20 mm and 38% of them had a cervical length <10 mm. Progesterone therapy was administered to all women after sample collection to reduce the risk of sPTB throughout gestation. Overall, 10 women (21.7%) delivered preterm (<37 weeks’ gestation): of these, three belonged to the group with extreme cervical shortening (<11 mm), six to the group with cervical length 11–20 mm, and only one to the group with cervical length 21–24 mm. The relationship between specific CSTs and PTB was not analyzed, as the study was not powered to evaluate this outcome.

As the first step of our analysis, we evaluated whether maternal age and/or the presence of GDM can affect cervical shortening. Appendix A shows that there were no significant differences in cervical length between women with GDM and women who did not develop GDM. Similarly, no significant correlations between GA at sampling or maternal age and cervical length were detected (Appendix A).

### 3.2. Vaginal Microbiota Composition

The vaginal samples were first classified into two categories based on *Lactobacillus* abundance (Figure 1A): (i) *Lactobacillus*-dominated (≥50% *Lactobacillus spp.*); and (ii) Lactobacillus-depleted microbiota (<10% *Lactobacillus* spp.). The *Lactobacillus*-dominated microbiota category was the most prevalent in our cohort (*n* = 40; 86.9%), while the *Lactobacillus*-depleted microbiota was only present in 6 women (13%). Spearman correlation analysis showed that the abundance of taxa belonging to the *Lactobacillus* genus was positively correlated with cervical length (Spearman coefficient 0.421; *p* = 0.01). Figure 1B shows that women with *Lactobacillus*-depleted microbiota had an extremely short cervix length (<10 mm) and significant differences were recorded by comparing the cervical length of these women with that of women with *Lactobacillus*-dominated microbiota. When the cervical length of these two groups was adjusted for maternal age or GA, no interaction between grouping variable (*Lactobacillus*-depleted/-dominated microbiota) and covariates was found by ANCOVA analysis (Appendix A), supporting the significance of differences in cervical length between women with *Lactobacillus*-depleted microbiota and women with *Lactobacillus*-dominated microbiota (Figure 1C,D).

### 3.3. Community State Type (CST) Distribution of Vaginal Microbiota

To deeper understand vaginal microbiota composition and to define microbial profiles associated with extreme cervical shortening, we stratified vaginal microbiotas into the five major vaginal community state types (CSTs), according to Ravel J et al. [11].

*L. crispatus*-dominated microbiota (CST-I) characterized 34.8% (*n* = 16) of pregnant women, *L. gasseri*-dominated microbiota (CST-II) was present in 13% (*n* = 6), *L. iners*-dominated microbiota (CST-III) in 32.6% (*n* = 15), *L. jensenii*-dominated microbiota (CST-V) in 6.5% (*n* = 3) of women (Table 2). *Lactobacillus*-depleted microbiota, defined as CST-IV was found in 13% (*n* = 6) of our patients (Table 2). Fisher’s Exact test showed a statistically significant difference in the distribution of CSTs in women with different cervical length (*p* = 0.007; Table 2).

To evaluate the distribution of cervical samples based on their microbial community composition, we performed PCoA ordination based on Bray-Curtis distances (Figure 2). This analysis showed that the distribution of the vaginal samples was clearly driven by CSTs (PERMANOVA, *p* =  0.001).

Moreover, the differential abundance of *Lactobacillus* and *Gardnerella* genera reflected the distribution among samples (Appendix A; PERMANOVA R2 = 0.741; *p* < 0.001). PERMANOVA analysis showed that no other variables, such as cervical length, or the presence of GDM, differed between CST types (Appendix A).

In accordance with the data shown in Figure 1B, OR calculation showed that vaginal microbiota of CST-IV type was the main risk factor for extreme cervical shortening (OR = 15 CI = 1.56–144.0; *p* = 0.019).

However, due to the high prevalence of women of Caucasian ethnicity in our cohort, CST-IV was present in only 13% of vaginal swabs, while *Lactobacillus*-dominated CSTs were largely represented in 87% of samples (Table 2). Thus, we decided to analyse only the *Lactobacillus*-dominated vaginal microbiota samples. We observed that in 7/15 women (46.7%) with an extremely short cervix (1–10 mm) the vaginal community was dominated by *L. iners* (Table 2). Statistical analyses indeed showed that, among *Lactobacillus*-dominated CSTs (CST I, CST II, CST III, CST V), women with CST III microbiota had the highest risk of having an extremely short cervix (OR = 6.4 CI 95% = 1.32–31.032; *p*-value = 0.024; Table 3). This suggests that CST-III might be the only community among that *Lactobacillus* dominated-species possibly involved in the mechanisms leading to severe cervical shortening.

We also measured MMP-8 concentration in supernatant of cervical samples, since this enzyme is known to compromise the epithelial barrier integrity [26] and was found to be strongly associated with cervical shortening [27]. We found an increased concentration of MMP-8 in women with CST III and CST IV microbiota compared to other CSTs groups (*Lactobacillus*-dominated community; CST I, CST II, and CST V). The difference, however, did not reach statistical significance (Appendix A).

### 3.4. Cervical Shortening, Microbial Diversity and Enrichment of Bacterial Taxa

When pregnant women were stratified in three subgroups based on cervical shortening (1–10 mm, 11–20 mm, 21–24 mm), microbial diversity seemed to increase with progressive cervical shortening, as shown by barplot configuration of microbiota composition in vaginal samples (Figure 3A). We used these groups of samples to further identify differential microbial profiles associated with cervical shortening. In particular, a progressive reduction of *Lactobacillus* spp. and an increase of *Gardnerella*, *Streptococcus*, *Enterococcus* and *Prevotella* genera were evident in groups with the shortest cervix (Figure 3A). Estimation of species richness (alpha diversity) was also measured (Appendix A). Although a trend toward an increased microbial diversity was evident in the 1–10 mm and 11–20 mm groups compared to the 21–24 mm group, no statistically significant differences in species richness were observed by ANOVA analysis.

The enrichment of selected bacterial taxa in groups of women with different cervical length was evaluated by LEfSe. By pairwise comparison of vaginal samples of women with cervical length 1–10 mm versus 21–24 mm (Figure 3B), we confirmed enrichment of *Lactobacillus* spp. in women with cervical length approaching the limit of 25 mm (21–24 mm). Taxa from the *Bifidobacteriaceae* family and, in particular, *Gardnerella* genus were enriched in women with a cervical length ranging from 11 to 20 mm, compared with women with a cervical length of 21–24 mm (Figure 3C). LEfSe, however, did not show discriminative bacterial profiles when samples of women with cervical length between 1 and 10 mm were compared to those of women with cervical length between 11 and 20 mm.

### 3.5. Gestational Diabetes Mellitus, Vaginal Microbiota Profile and Cervical Shortening

In our cohort, 21.7% (10/46) of pregnant women were diagnosed with GDM (Table 1), a metabolic disorder that may affect the composition of the vaginal microbiota [19]. CSTs distribution of microbiota from women with GDM revealed 3 CST-I, 3 CST-III, 3 CST IV, and 1 CST-V. No difference in the distribution of samples based on GDM was highlighted by the PCoA ordination (PERMANOVA; Appendix A). Moreover, as indicated above, Figure 1A shows no statistically significant differences in the cervical length between women with GDM and those not affected.

Although the number of women with GDM was limited, we noted that alpha diversity indexes (observed OTUs and Chao I) were significantly higher in women experiencing GDM compared with non-diabetic women (Figure 4A).

To investigate the reasons for this increase in bacterial richness in more depth, we compared vaginal microbial profiles of women experiencing GDM during pregnancy with those of non-diabetic women. LEfSe analysis revealed significant enrichment of taxa in association with GDM, such as *Fusobacterium*, *Mobiluncus*, *Prevotella*, *Brevibacterium*, and taxa from the families of *Enterobacteriaceae* (*Campylobacter*, *Haemophilus*), *Aerococcaceae*, *Sutterellaceae* and *Lachnospiraceae* (Figure 4B). When women with GDM were considered as a subclass stratified based on cervical shortening, LEfSe analysis did not reveal significant enrichment of bacterial taxa, indicating no association between specific microbiota profiles and short cervix in women with GDM.

## 4. Discussion

In our study, microbiota analysis of vaginal fluids was performed in a selected cohort of pregnant women with cervical shortening during the second or early third trimester of pregnancy, to identify vaginal communities associated with “extreme” cervical shortening (1–10 mm), a high-risk factor for spontaneous preterm birth [1,2,3]. A cervical length shorter than 10 mm is considered abnormal (below the 5th or 10th percentile for gestational age) even at 28–32 weeks’ gestation [1,20]. Iams et al. [1] reported that the relative risk of PTB increased as the length of the cervix decreased: they observed that the RR for PTB was 9.49 (95% CI 5.95–15.15) for lengths at or below the 5th percentile at 24 weeks (22 mm), and 13.99 (95% CI 7.89 to 24.78) for lengths at or below the 1st percentile (13 mm), compared with those above the 75th percentile. At 28 weeks, the corresponding relative risks for preterm delivery were 13.88 and 24.94.

In agreement with Gerson et al. [8], we found that *Lactobacillus*-abundance was positively correlated with cervical length. In contrast, *Lactobacillus*-depleted communities, which define the microbiota commonly named CST-IV [11], were significantly associated with increased odds of extreme cervical shortening (OR = 15 CI = 1.56–144; *p* = 0.019). Taking into account the higher stability of vaginal microbiota during pregnancy compared to non-pregnant status [17], these data reinforce the concept that *Lactobacillus*-depleted communities leading to vaginal dysbiosis are a risk factor for cervical insufficiency and remodeling of the cervix during pregnancy. *Lactobacillus*-dominated communities of the vagina are known to inhibit the adhesion and proliferation of opportunistic and primary pathogens [28] through multiple mechanisms including the production of antimicrobial compounds, such as hydrogen peroxide, lactic acid and/or bacteriocins, acting as a biosurfactant on the vaginal epithelium [10].

The occurrence of communities with low proportions or no detectable *Lactobacillus* spp. are relatively uncommon in the vaginal environment of non-pregnant white Caucasian women (10.3%) or Asian women (19.8%) compared with Hispanic (38.1%) and Black (40.4%) women [11].

In our cohort, white Caucasian women represented 96% of cases, Asian 2.1%, and only one woman was from North-Africa (Arabian ethnicity). *Lactobacillus*-dominated communities were present in 86% of vaginal swabs. Although our data confirmed the association between CST-IV and extremely short cervix [8], we also evaluated whether other microbial communities, more represented in white Caucasian women, could also be associated with the risk of an extremely short cervix. Our results showed that about half of women with *L. iners*-dominated communities (CST-III) had an extremely short cervix at the time of sampling, suggesting that *L. iners* may play a role in the mechanisms of cervical shortening and remodeling during pregnancy. Compared to the other species of Lactobacilli evolutionary adapted in the vaginal environment, *L. iners* is the species with the lowest ability to contrast infections from external pathogens or pathobionts. *L. iners* produces D-lactate instead of L-lactate, low amounts of antimicrobial peptides, and has reduced ability to bind epithelial cells [12]. For these reasons, *L. iners* has reduced ability to prevent the enrichment of *Gardnerella* and other bacteria causing bacterial vaginosis and it is better adapted to vaginal dysbiosis-associated conditions, such as an elevated pH and the presence of polymicrobial communities [10,12,29,30].

*L. iners* has been suggested as a marker of microbial imbalance leading to bacterial vaginosis [31]. Moreover, it was reported that *L. iners* increases ectocervical and endocervical permeability, suggesting that this bacterial species is less active in modulating inflammatory processes that could have negative consequences on cervical length during pregnancy [31].

Kindinger et al. [32] found a significant positive association between *L. iners*-dominated communities (CST-III) and the occurrence of spontaneous pre-term birth in a cohort of predominantly Caucasian and Asian women. In our study, the *L. iners*-dominated community was the only *Lactobacillus*-enriched community significantly associated with an extremely short cervix (1–10 mm; OR = 6.4; *p* = 0.02) suggesting that, besides CST-IV, CST-III may serve as a marker of increased risk of extreme cervical shortening, in particular in women of Caucasian ethnicity.

Finally, it is known that insulin resistance, weight gain and increased inflammation in women developing GDM may play a role in favoring adaptation of microbial communities that are different from those of non-diabetic women [19]. An abundance of potentially pathogenic bacteria and an increase of inflammatory cytokines expression have been described in the vaginal microbiome of women with GDM. We thus investigated whether, in our cohort, women experiencing GDM during pregnancy have a different vaginal microbial profile compared to women who did not develop GDM. In accordance with Cortez et al. [19], we found that the vaginal microbial profiles of women with GDM were enriched of bacterial taxa abundant in vaginal dysbiosis or associated with a viral infection, inflammation or epithelial adhesiveness [13,33] compared to non-diabetic women. Despite these results, we could not associate GDM with extreme cervical shortening. Further studies in a larger cohort of pregnant women are needed to define whether GDM or other complications during the pregnancy are involved in the mechanism leading to cervical shortening.

Strengths of the present study include the cross-sectional design and the selection of a group of mostly Caucasian women, who generally have a vaginal microbiota enriched with *Lactobacillus* spp. compared to other ethnicities, to investigate the association between microbial profiles/CSTs and a shortened cervical length. Furthermore, vaginal sampling was performed before any mitigative or therapeutic measures in order to limit the amount of confounding factors.

This study presents some limitations: (i) only one sample of vaginal fluid was collected from each woman for vaginal microbiota investigation. This may represent a limitation, as it does not allow for evaluating the dynamics of the microbiota with progressive cervical shortening during pregnancy. However, some evidence indicated that, during physiologic pregnancy, vaginal microbiota is more stable compared to non-pregnant women [18]; (ii) assessing the global vaginal microbiota community by 16S rRNA gene-based amplicon sequencing limits the evaluation of every single bacterial contribution to the mechanisms leading to cervical shortening; (iii) we could not draw any conclusion on the relationship between specific CSTs and PTB, as the study was not powered to evaluate this outcome. Moreover, the treatment of most women with progesterone therapy limited the ability to evaluate such association; (iv) the study did not include a control group of women with cervical length >25 mm, which would have allowed comparison of microbial profiles according to cervical length; (v): the exclusion of patients who had a pessary placed for prevention of PTB could represent a selection bias. At the time of the study at our institution, the cervical pessary was placed, with specific indications, during hospital admission and then the patients received follow-up at our preterm birth clinic. Therefore, we did not include this subset of patients, as the impact that the pessary might have on the vaginal microbiota is not well known.

This study showed that CST-IV is a risk factor for extreme cervical shortening in Caucasian women. *L. iners*-dominated community (CST III), a type of vaginal microbiota much more common in white Caucasian women, was identified as an additional risk factor for extreme cervical shortening. Future studies exploring the microbial contribution to the mechanisms leading to severe cervical shortening will be crucial in predicting susceptibility to sPTB.

## Figures and Tables

**Figure 1 jcm-09-03621-f001:**
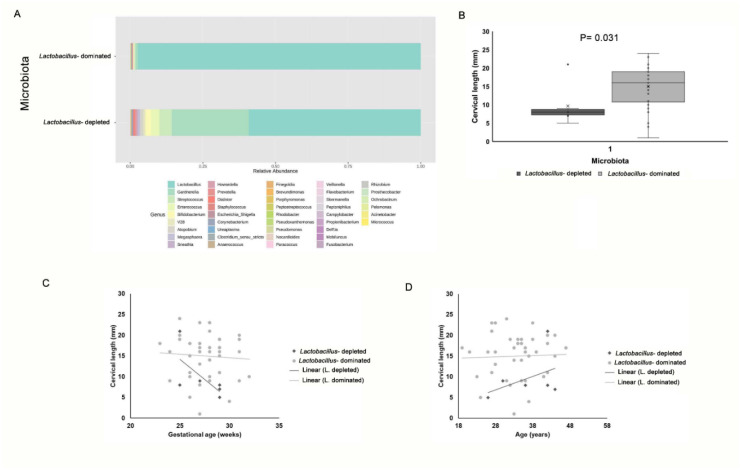
Survey of vaginal microbiota and correlation with cervical length. (**A**) Overview of vaginal microbiota grouped by *Lactobacillus*-depleted and *Lactobacillus*-dominated microbiota types. In each barplot, the percentage of relative abundances at the genus level is showed. (**B**) Differences in cervical length between women with *Lactobacillus*-depleted microbiota and women with *Lactobacillus*-dominated microbiota. Data are presented as box and whisker plots, with boxes extending from the 25th to 75th percentile and horizontal lines representing the median. Whiskers extend 1.5 times the interquartile range from the 25th and 75th percentile. Statistical analysis was performed by Mann–Whitney assay. *p*-value < 0.05 was considered as significant. (**C**,**D**) Analysis of covariance (ANCOVA) with grouping variables and covariates (**C**) age and (**D**) gestational age at sampling. Scatter plot with regression lines for the two groups (*Lactobacillus*-depleted/*Lactobacillus*-dominated microbiota).

**Figure 2 jcm-09-03621-f002:**
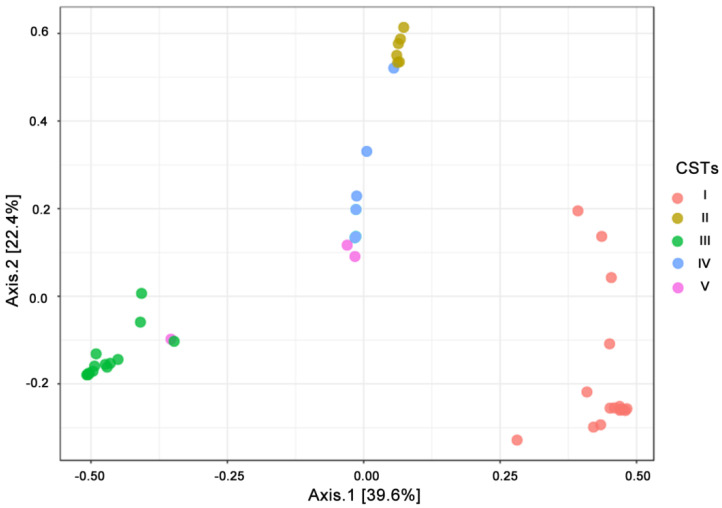
Beta diversity measure. Principal Coordinate Analysis (PCoA) ordination based on Bray Curtis dissimilarities correlated with community state types (CSTs) (permutational multivariate analysis of variance (PERMANOVA) 999 permutations; R2 = 0.740 *p*-value < 0.001). Samples belonging to different CSTs are indicated with different colour dots.

**Figure 3 jcm-09-03621-f003:**
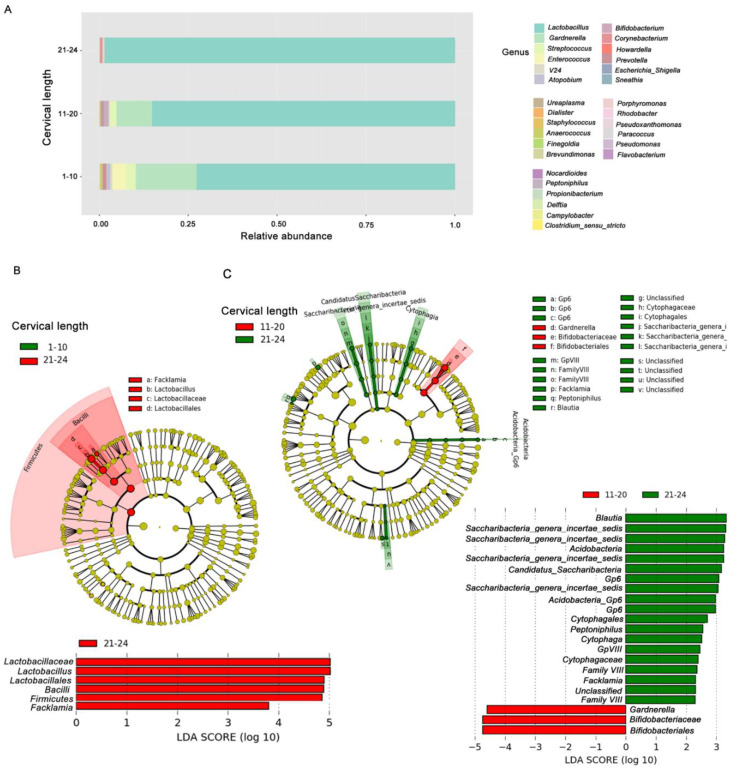
Microbiome profiles in vaginal samples according to cervical shortening categorization. (**A**) Barplot of vaginal swab samples was stratified based on cervical shortening (1–10 mm, 11–20 mm, 21–24 mm). The percentage of bacterial relative abundances (average) at the genus level is showed. (**B**,**C**) Metagenomic biomarker discovery by linear discriminant analysis effect size (LEfSe) analysis. Comparison of enriched taxa between vaginal samples of women with cervix length (**B**) 1–10 mm vs. 21–24 mm, and (**C**) 11–20 mm vs. 21–24 mm. Results indicated the statistically significant taxa enrichment among groups (Alpha value = 0.05 for the factorial Kruskal–Wallis test among classes). The threshold for the logarithmic LDA score was 2.0.

**Figure 4 jcm-09-03621-f004:**
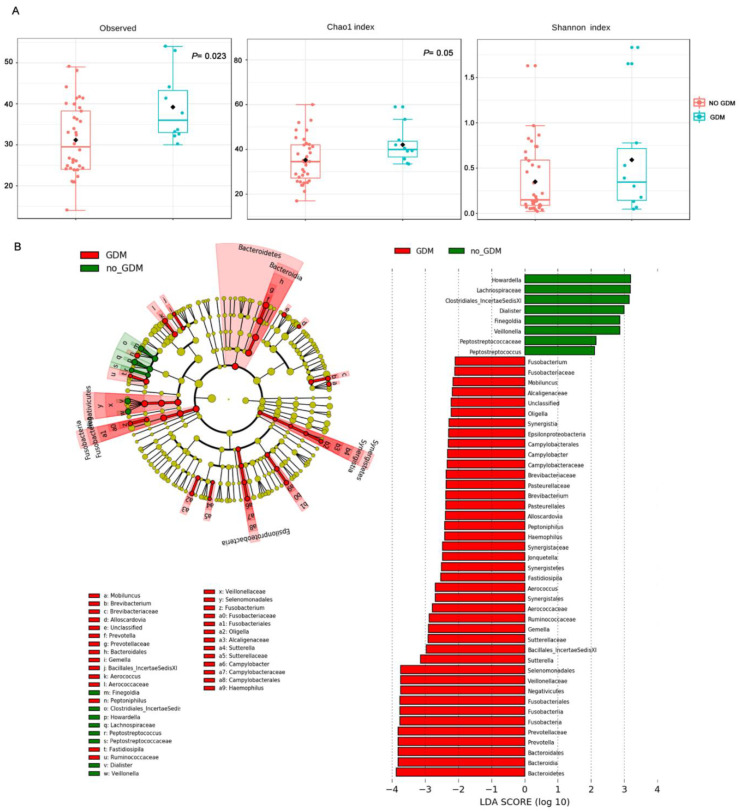
Vaginal microbiome profiles of women experiencing GDM during the pregnancy. (**A**) Alpha diversity (observed operational taxonomic units (OTUs), Chao I and Shannon indexes; *p*-values by Kruskal–Wallis test). (**B**) LEfSe analysis. Comparison of enriched taxa between vaginal samples of women with GDM and normal glucose-tolerant women. Results indicated the statistically significant taxa enrichment among groups (Alpha value = 0.05 for the factorial Kruskal–Wallis test among classes). The threshold for the logarithmic LDA score was 2.0.

**Table 1 jcm-09-03621-t001:** Patient demographic and clinical information.

	All Women, *n* (%)	Stratification by Cervical Length (<25 mm)			*p*-Value (Chi Square Test)
		1–10 mm	11–20 mm	21–24 mm	
N. of enrolled women at risk of sPTB	46 (100%)	15 (32.6%)	25 (54.3%)	6 (13%)	
**Ethnicity**					
Caucasian	44 (95.7%)	15 (100%)	23 (92%)	6 (100%)	
Asian	1 (2.2%)	0 (0%)	1 (4%)	0 (0%)	
North-African (Morocco)	1 (2.2%)	0 (0%)	1 (4%)	0 (0%)	
**Age at sampling (years);**					
mean ± SD	34.2 ± 6.7	34.1 ± 6.5	34.7 ± 7.1	32.2 ± 6.4	
**Gestational age at sampling (weeks);**					
mean ± SD	27.6 ± 2	27.9 ± 2.2	27.6 ± 2.1	26.8 ± 1.5	
**Pregnancy complications**					
Vaginal infection ^1^	11 (23.9%)	5 (33.3%)	4 (16%)	1 (16.6%)	0.415
GDM	10 (21.7%)	1 (6.66%)	7 (28%)	2 (33.3%)	0.217

^1^ Vaginal infections included yeast infection and bacterial vaginosis (such as *Streptococcus, Gardnerella, Ureaplasma, Klebsiella* and *Citrobacter*). These infections were diagnosed at a later gestational age than enrolment and sample collection, when the patient reported vaginal symptoms. sPTB, Spontaneous preterm birth; GDM, gestational diabetes mellitus

**Table 2 jcm-09-03621-t002:** Distribution of CSTs in all recruited women, according to cervical length. We differentiated the women with a very short cervix (1–10 mm) from the others (11–24 mm).

		Cervical Length		
	All Women	1–10 mm	11–24 mm	*p*-Value (Fisher’s Exact Test)
**Total N**	46	15	31	
**CST I**	16 (34.8%)	3 (20.0%)	13 (41.9%)	0.007
**CST II**	6 (13.0%)	0 (0.0%)	6 (19.4%)	
**CST III**	15 (32.6%)	7 (46.7%)	8 (25.8%)	
**CST IV**	6 (13.0%)	5 (33.3%)	1 (3.2%)	
**CST V**	2 (6.5%)	0 (0.0%)	3 (9.7%)	

**Table 3 jcm-09-03621-t003:** Association of CSTs and cervical shortening categorization.

Cervical Length						
Vaginal Microbial Community	1–10 mm	11–24 mm	*p*-Value (Fisher’s Exact Test)	OR	95% CI	*p*-Value OR
***L. iners*-dominated community (CST III)**	7 (46.7%)	8 (25.8%)	0.191	2.52	0.69–9.18	0.16
**other CSTs**	8 (53.3%)	23 (74.2%)				
***L. iners*-dominated community (CST III)**	7 (70.0%)	8 (26.7%)	0.024	6.417	1.327–31.032	0.021
**other *Lactobacillus* spp.-dominated community (CST I, CST II, CST V)**	3 (30.0%)	22 (73.3%)				

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
