# Peer review of "Identification of Vaginal Microbial Communities Associated with Extreme Cervical Shortening in Pregnant Women"

_jcm, 2020, doi:10.3390/jcm9113621_

Round 1
Reviewer 1 Report
Investigating the role of vaginal microbiota in cervical shortening and subsequent PTB is a worthwhile pursuit.
The sampling, extraction and microbiome data analysis are sound
There are a few methodological issues for which further metadata would be helpful.
Exclusion criteria included a Hx of PTB, PPROM, Contractions and fetal anomalies but it is not clear if any of the subjects had previous cervical treatment which would result in a short cervix, I assume not but this should be stated in the manuscript.
The results identify that there is an association between a short cervix and CST IV / CST III. This single time-point represents a snapshot and it is not possible to decipher if there is evolving cervical shortening as a result of the vaginal microbiome or if the cervix was always short regardless of the microbiome. The conclusion that "CST-III may also be involved in cervical shortening" has not been demonstrated by the data presented.
The methods state that supernatant from the samples was also stored, quantifying markers of inflammation and correlating this to CST would add strength to the results.
The sample size is small and as stated by the authors was not powered to look at PTB outcomes. This impacts the clinical relevance of the study findings. Did any of the women with short or extremely short cervixes deliver preterm?
The data presented for women with GDM is interesting but the small numbers (n=10) preclude detailed analysis and conclusions from being made.
Reviewer 2 Report
Line 37-38 - Please edit to read: 'Premature cervical remodeling, shortening and dilation of the cervix are known risk factors for sPTB, with the notion that the shorter the cervix, the higher the risk of sPTB.'
Line 40 - Add a reference for 'progesterone deficiency'
Line 41- Stating that 'Intra-amniotic infection' leads to cervical shortening is actually quite controversial, since many studies of preterm birth have found that amniotic cavity to be sterile. Since your study only focuses on changes in the vaginal microbiota, I would reword that to say ' In addition to congenital disorders, genetic syndromes and progesterone deficiency, local inflammation secondary to changes in the cervico vaginal microbiome are another mechanism that has been proposed to cause cervical shortening.
Line 54-55 - I would specify if you mean in pregnant or non-pregnant women- Reference 15 only speaks to a pregnant cohort, but references 9 and 11 are non-pregnant women. Please reread references and use appropriately.
Line 55-66 - These two paragraphs need reworking, because they are not organized, and the authors are not making coherent points. I think they are trying to say the following:
Point 1, CST-IV is composed of mainly anaerobic bacteria, and is more prevalent in Black non pregnant women than Caucasian non-pregnant women.
Point 2: In pregnant women, CST IV has been associated with a higher risk of sPTB and short cervix, while CST I or Lactobacillus dominated CSTs has been associated with more favorable pregnancy outcomes. Please be aware Reference 15 and Reference 22 used the same cohort, and so should be quoted together.
Point 3: Metabolic disorders of pregnancy such as GDM may affect the composition of the vaginal microbiome and pregnancy outcomes, as they have been shown to increase inflammatory cytokines.
Line 69- The authors did not explain why they chose to stratify by 'extreme cervical shortening,' and should do somewhere in the introduction. Also, is 'extreme cervical shortening' a clinical definition? And why did they chose that particular length, ie has 10 mm been shown to correlate with worse pregnancy outcomes over 25 mm?
Methods
Line 74- Do women at this hospital all receive a transvaginal cervical length as part of a protocol? Or were these women recruited specifically for the study?
Line 75- Please give the specific gestational ages from when to when in gestation the scans were performed - ie from 16w0d - 25w0d, etc.
Lines 74 -98- The exclusion/inclusion criteria are a little confusing, and this whole paragraph should be edited for clarity. The authors should clearly state how many women were approached, how many were excluded (and why) and how many women were included in the final cohort. This can be done in a figure, but it should be done somewhere. There is always bias inherent in creating a cohort, but the authors are so vague in their descriptions, it is hard to tell if a) all pregnant women who did not meet the exclusion criteria were approached, or just a subset (ie its unclear whether this hospital has a universal cervical length screening protocol in place or these women were randomly selected by the study authors), and b) these women had to refuse an intervention (pessary) in order to be included (which will probably add in significant bias to your sample). Also, I would assume the authors only included singleton pregnancies, but that should be made clear.
Results
Line 160 - The gestational age is very late for cervical length screening, as the vast majority of the studies done showing that short cervix correlates to a risk of sPTB were done at the midtrimester, and so it is a big leap to say a short cervix at 20 weeks carries the same risk as a short cervix at 28 weeks, since most perinatal units do not routinely screen for transvaginal cervical lengths that late in pregnancy unless women present with symptoms (which is not the meant to be the case in this paper). Most protocols in the United States at least only perform transvaginal cervical length screening until 24-25 weeks, and so again, the authors need to better describe their protocol (ie why were these women getting transvaginal cervical lengths so late in pregnancy)?
Line 176 - I would not refer to 'dysbiotic' as that is too generalized a term- I would refer to it as Lactobacillus deficient and define that as the authors have (ie < 10% of Lactobacillus spp).
Line 183- I would avoid using terms such as 'healthy'- since we don't yet understand what healthy vs dysbiotic- just that we see differences.
Figure 1 - Again, would not use 'Healthy' vs 'Dysbiotic' - just Lactobacillus dominant and deficient
Line 268 - The authors should define how GDM is diagnosed, how long these women had been affected by GDM, and whether they were diet or medication controlled.
Line 280- Just say 'non diabetic women'- 'normally glucose tolerant' is incorrect English.
Discussion-Lines 297-299 - Again, I cannot entirely agree with the authors conclusion about their cohort, since they have not provided good evidence to say that a short cervix at 28-29 weeks correlates with an increased risk of sPTB. A more precisely defined cohort is needed, because otherwise it will be difficult to correlate these results to clinical practice, where most units only measure the cervix until 24 or 25 weeks (as is the case of Reference 22- the authors cite Ref 21 here, but they mean Ref 22).
313- Please reword 'relative stablization' - imprecise English. And again, define dysbiotic.
Round 2
Reviewer 1 Report
The authors have addressed all the issues raised in the original review.
This study provides further evidence of the association between vaginal microbial composition and cervical shortening which is a significant risk factor for subsequent PPROM/ sPTB. Within the limitations of the study described by the authors. Data relating to those women with GDM is less convincing given the small sample size but may represent preliminary data to support larger investigation into the affect of systemic metabolic dysfunction such as that seen in GDM/ T2DM and T1DM upon the vaginal microbiome and Activation of inflammation within the vagina
Author Response
We thank the reviewer for his/her valuable and insightful comments. We would like to consider the results on GDM and microbiota profiles as preliminary data. In the Results section (lines 321-323), we reported as follows: “Although the number of women with GDM was limited, we noted that alpha diversity indexes (Observed OTUs and Chao I) were significantly higher in women experiencing GDM compared with non-diabetic women”. Moreover, in the Discussion section, we have already indicated that further studies on a larger cohort of women with GDM are needed to confirm the data presented in this manuscript (lines 399-401).
Reviewer 2 Report
Abstract:
Line 17 - Vaginal microbiota play a critical role in pregnancy.
Line 17-19 - Would reword slightly, English is unclear. Maybe: 'Bacteria from the Lactobacillus spp. genus are thought to maintain immune homeostasis and modulate the inflammatory responses against the pathogens implicated in cervical shortening, one of the risk factors for spontaneous preterm birth.'
Line 20 - Should be corrected as follows: We studied the vaginal microbiota in 46 pregnant women of predominantly Caucasian ethnicity diagnosed with short cervix (<25 mm), and identified microbial communities associated with extreme cervical shortening (≤10 mm).
Introduction: Much improved, no further comments.
Methods: Much improved, although would still like to know the total number of women approached, and how many were excluded (if known).
Results/Discussion: Improved, no further comments other than the authors should have a native speaker review (if possible) to correct a few minor grammar mistakes.
Author Response
Response 1 (Abstract): Regarding the "microbiota", we have found that it is generally used as a singular noun also in other papers, and it seems to be correct. Therefore, we choose to leave the verb “plays”.
Responses 2-3 (Abstract): We thank the reviewer for his/her comments on our revised manuscript and for the useful suggestions. We have made the suggested changes (lines 17, 17-20, and 205-207).
Response 4 (Methods): During the study period (2014-18), 174 potential study participants who met the inclusion criteria were identified. Of these, 128 were excluded because they presented one or more of the exclusion criteria, leaving 46 subjects available for enrollment. We have now included this data in the manuscript (Results section).
Response 5 (Results/Discussion): We requested the English revision of a native speaker. We hope that the language revision improved the quality of the manuscript.